# Lipid Myopathies

**DOI:** 10.3390/jcm7120472

**Published:** 2018-11-23

**Authors:** Elena Maria Pennisi, Matteo Garibaldi, Giovanni Antonini

**Affiliations:** 1Unit of Neuromuscular Disorders, Neurology, San Filippo Neri Hospital, 00135 Rome, Italy; 2Unit of Neuromuscular Diseases, Department of Neurology, Mental Health and Sensory Organs (NESMOS), SAPIENZA University of Rome, Sant’ Andrea Hospital, 00189 Rome, Italy; matteo.garibaldi@uniroma1.it (M.G.); giovanni.antonini@uniroma1.it (G.A.)

**Keywords:** lipid myopathies, lipid storage disease, muscle lipidosis, lipid metabolism disorders, beta-oxidation defects, FAO defect, metabolic myopathies

## Abstract

Disorders of lipid metabolism affect several tissues, including skeletal and cardiac muscle tissues. Lipid myopathies (LM) are rare multi-systemic diseases, which most often are due to genetic defects. Clinically, LM can have acute or chronic clinical presentation. Disease onset can occur in all ages, from early stages of life to late-adult onset, showing with a wide spectrum of clinical symptoms. Muscular involvement can be fluctuant or stable and can manifest as fatigue, exercise intolerance and muscular weakness. Muscular atrophy is rarely present. Acute muscular exacerbations, resulting in rhabdomyolysis crisis are triggered by several factors. Several classifications of lipid myopathies have been proposed, based on clinical involvement, biochemical defect or histopathological findings. Herein, we propose a full revision of all the main clinical entities of lipid metabolism disorders with a muscle involvement, also including some those disorders of fatty acid oxidation (FAO) with muscular symptoms not included among previous lipid myopathies classifications.

## 1. Introduction

Lipid myopathies (LM) are a group of muscular diseases with onset in all ages, due in most cases to enzymatic errors of lipid metabolism. LM often, but not always, are characterized by lipid storage in muscle biopsy. Clinically LM can show stable and progressive weakness or episodic asthenia with rhabdomyolysis crisis, triggered by metabolic stress, fever, physical exercise.

Lipids are essential for the structural and functional maintenance of cells. Fatty acids (FA), the basic molecules of lipids, play an essential role in energy production in skeletal and cardiac muscle cells, in particular during fasting and prolonged exercise. Acute lipid oversupply produces inhibition of glucose oxidation, and mitochondria preferentially switch from carbohydrate to FA utilization, depicting the high degree of metabolic flexibility in skeletal muscle. The variation in FA entrance in the fiber muscle is able to modulate FA oxidation rate, indicating a high level of the metabolic regulation. Fatty acids are metabolized by muscle fibers through a complex process consisting in 4 principal steps (Figure 1) that require several specific proteins: (1) entrance of fatty acids into the cytoplasm through proteins of the sarcoplasmic membrane; (2) storage or breakdown of FA of lipid droplets; (3) transportation of FA into the mitochondria, from external to internal mitochondrial membrane; (4) FA release in mitochondria matrix to enter in beta-oxidation. The carnitine protein is mainly responsible for the FA entrance trough cell membrane. The FA are stored in cytoplasm as triglycerides (TG) in lipid droplets (LD), that are organelles derived from the endoplasmic reticulum, surrounded by phospholipid and regulatory enzymes [1]. TG in LD can be subsequently hydrolyzed to FA by a subset of lipases, namely adipose triglyceride lipase (ATGL), activated by the coenzyme Comparative Gene Identification-58 (CGI58), hormone sensitive lipase (HSL) and monoglyceride lipase (MGL), in a process called lipolysis [2]. FA entry the mitochondria thanks to the action of Carnitine Palmitoyl Transferase I (CPT I), Mitochondrial Trifunctional Protein (MTP), Carnitine- Acylcarnitine Translocase (CACT) and Carnitine Palmitoyl Transferase II (CPT II). The FA are metabolized in mitochondria by Electron-Transfer Flavoprotein (ETF) and beta-oxidation enzymes. Any change in these different steps can cause a lipid myopathy which, similarly to other metabolic myopathies, can be clinically characterized by exercise intolerance with fluctuant or fixed weakness, rhabdomyolysis and myoglobinuria (red brown colored urine), myalgias, eventually associated to muscle atrophy and involvement of other organs (liver, heart and central nervous system). Most lipid diseases are caused by a gene defect, they are more rarely produced by a toxic cause or by an unbalanced diet. In the forms with fixed myopathy, dosage of Creatin Kinase (CK) ranges from 2 to 5 times the normal value. Conversely the lipid myopathies that occur with crisis, rhabdomyolysis and myoglobinuria can cause, during the acute onset, the rise of CK to up to several thousands of units, provoking the risk of renal failure. Excess of lipid in the cytoplasm of muscle fibers can be visible, in the majority of the cases, in muscle biopsy, with Oil Red O (O.R.O.) or Sudan Black staining on criosections. Dystrophic changes are generally absent. Type I fibers are more affected than type II fibers, because of their major utilization of lipids in cellular metabolism. A normal muscular biopsy does not rule out lipid myopathy, because some of these diseases do not cause excessive lipid storage, as in CPTII deficit, so the term of lipid storage myopathies could be inappropriate to define all type of lipid myopathies. Several biochemical investigations based on dosage of intermediates products of lipid catabolism (acylcarnitines profile and urinary organic acids) could help in the diagnostic work-up of lipid myopathies before going to the genetic confirmation [3,4,5]. Cellular functional studies can contribute to clarifying the residual enzymatic activity of one specific enzyme in particular cases [6]. The role of MRI, a largely used method of studying myopathies, is still not completely investigated in lipid myopathies but some papers have been published for specific myopathies, as neutral lipid storage disease (NLSD), MTP and Very-Long-Chain Acyl-CoA Dehydrogenase (VLCAD) [7,8,9,10].

Evidence-based studies concerning the therapy of lipid myopathies are very scarce. Most of the recommendations for the treatment of lipid myopathies are shared with all types of LM, the avoidance of precipitating factors, such as prolonged fasting, aerobic exercise (>30 min), infections and cold exposure should be recommended in all FA disorders. Fundamental saving-life procedures for episodes of rhabdomyolysis of any cause, comprehend intensive saline and glucose solutions administration and monitoring of electrolytes, kidney, and cardiac functions. In other diseases dietary supplementation of deficient molecules, such as with carnitine in Primary Carnitine Deficiency (PCD), is sufficient to restore the previous conditions.

In this review only the myopathic forms of lipid inborn errors will be revisited, while the forms without myopathic involvement, including defects of Long-Chain Fatty Acids Uptake (LCFAUD), Acylglycerol-3-Phosphate O-Acyltransferase 2 (AGPAT2), Diacylglycerol O-acyltransferase 1 (DGAT1), Hormone-sensitive lipase (HSL), carnitine palmitoyl transferase I (CPTI) and 3-hydroxy-3-methylglutaryl-coa lyase deficiency (HMG-CoA lyase), will not be discussed. The spectrum of LM is expanding with the knowledge of new molecules involved in FA metabolism. In the past, some reviews have been proposed with different disease classification based on clinical, biochemical or morphological features, but because of the complexity of the topic, a definitive classification remains in progress [10,11]. In our review we propose a subdivision of lipid myopathies in three groups based on mutated genes, following the enzymatic cascade that brings from the entrance of lipid into the cell, to the cytoplasmic arrangement and the beta-oxidation. We have chosen to insert in the lipid myopathies classification all the diseases caused by mutations of genes encoding for enzymes of lipid metabolism, that present myopathic symptoms, including those forms in which other systemic symptoms can be present in the clinical picture, but in which the muscular involvement has been well documented. Causative genes, symptoms, laboratory findings and therapy data are summarized in Table 1.

## 2. Diseases

### 2.1. Primary Carnitine Deficiency (PCD)

Primary Carnitine Deficiency (PCD) is characterized by a low concentration of plasma and tissue carnitine [12,13] due to recessive mutations in *Solute Carrier Family 22 Member 5* (*SLC22A5)* gene which encodes Organic Cation/Carnitine Transporter 2 (OCTN2) [14]. Many different mutations have been reported resulting in variable residual activity of OCTN2 transporter [3] and frequency of carriers is estimated around 1% in Japan [15]. OCTN2 function is to actively transfer carnitine across plasma membrane into cytoplasm. OCTN2 is expressed in several tissues, including muscle, heart and kidney, but not liver. Defects impairing OCTN2 function lead to a decreased renal threshold for carnitine reabsorption with urinary loss and failure of intracellular accumulation, resulting in marked reduction of total carnitine (<5% normal) with normal esterified fraction [12].

#### 2.1.1. Clinical Presentation

PCD could manifest with a wide spectrum of clinical symptoms, ranging from asymptomatic patients, episodic exertional rhabdomyolysis, hypertrophic or dilated early onset cardiomyopathy, generalized muscle weakness, to severe infantile Reye-like syndrome mainly characterized by recurrent hypoglycemic hypoketotic encephalopathy. Even if asymptomatic patients seems to have average higher levels of residual carnitine transport activity compared to symptomatic patients [16], no clear correlation has been assessed between residual uptake activity and severity of clinical presentation, suggesting that phenotypic variability in PCD could be related to epigenetic or exogenous factors exacerbating carnitine deficiency [17].

#### 2.1.2. Laboratory Data and Diagnosis

Muscle biopsy in PCD frequently shows a massive lipidosis, particularly evident in type 1 fibers, whereas type 2 could result atrophic [18]. Electron microscopy shows that lipid droplets are frequently close to enlarged, but structurally normal, mitochondria. Biochemical investigations reveal very low (<5% of normal) concentration of plasma carnitine with normal esterified fractions and high carnitine loss in urine [19]. Secondary reduction of total carnitine (<50% of normal) with increased acylcarnitines (>50% esterified; normal: 10–25% in fed state and 30–50% in fasting state) can be observed in several genetic and acquired conditions (Table 2). Diagnosis can be confirmed demonstrating the reduced carnitine uptake in lymphocytes and fibroblasts [12] or by analysis of *SLC22A5* gene [14]. 

#### 2.1.3. Therapy

Supplementation treatment with of high-dose of oral carnitine (100 mg/Kg/day) in four daily doses, dramatically improves myopathy and cardiomyopathy during the first weeks and prevents the development of phenotype in neonates [20]. Assumption of pivolic acid with antibiotics can deteriorate conditions of patients with this disease.

### 2.2. Neutral Lipid Storage Disease (NLSD)

The enzymatic inborn errors affecting the lipase ATGL (OMIM 609059) and his coactivator CGI58 (OMIM 604780) cause neutral lipid storage diseases (NLSD) which have myopathy, ichthyosis and cardiomyopathy as main symptoms. NLSD are two rare autosomal recessive disorders characterized by the excessive, non-lysosomal accumulation of neutral lipids in multiple tissues [20,21]. In these disorders the defects of enzymes result in muscle atrophy, cardiomyopathy, lipid accumulation and dysfunction of several internal organs as well as presence of ichthyosis. To date more or less 60 cases of NLSD-type M and 130 of NLSD type I are diagnosed around the world, in the Mediterranean area, Japan, China, the Arabian peninsula, and South America. In the general healthy working population, 2.6% of the individuals were found to harbor rare missense and nonsense mutations in ATGL [22], based on this, it is possible that these diseases are underdiagnosed in the neuromuscular patients population.

#### 2.2.1. Clinical Presentation

Structural defects of the *Patatin Like Phospholipase Domain Containing 2* (*PNPLA2*) gene mainly cause heart and skeletal muscle symptoms, whereas mutations in CGI58, the activator of PNPLA2, cause also ichthyosis and severe hepatic symptoms, associated with myopathic symptoms. The mechanisms leading to muscle damage remain largely unknown. In NLSD lipids accumulate in several tissues such as skin, skeletal muscle, liver, heart, thyroid, pancreas, central nervous system, and leukocytes. The cardiopathy in NLSD-M can produce severe cardiac failure in patients, so they must be treated with cardiac electric devices or transplant [23,24]. Severe liver failure can induce death in NLSD-I. The myopathy is fixe, with muscular hypotrophy, generally bulbar and respiratory muscles are not involved in both forms. Rarely double-sided cataracts, growth retardation, ataxia, bilateral sensorineural hearing loss, and/or mental retardation are reported [24]. The clinical presentation and evolution can be different in homozygotes born in the same family, so it is thought that other factors besides the genetic ones can influence the expression of the disease [24,25].

#### 2.2.2. Laboratory Data and Diagnosis

Muscle tissue histology in both NLSD-M and NLSD-I patients revealed mild atrophy and lipid vacuolization of fibers. No increase in connective or adipose tissue is present. No cellular infiltrates or significant necrosis are detected. Lipid droplets in the cytoplasm of muscle fibers are detected by means of optic microscopy in 93% of the muscle biopsies [25,26] and were positive for O.R.O. or Sudan Black staining (Figure 2). Muscle MRI has a role in diagnostic assessment [8]. The whole-body study showed involvement of muscles gluteus minimus, semimembranosus, soleus and gastrocnemius medialis in the lower limbs and infraspinatus in the upper limbs were the most affected muscles, with a specific pattern of muscle involvement with “patchy” areas of fatty replacement, while mm. gracilis, sartorius, subscapularis, pectoralis, triceps brachii and sternocleidomastoid were spared. However the most important characteristic of NLSD is the presence of lipid-containing vacuoles in white blood cells in blood peripheral smear (the so called “Jordans’ anomaly” (Figure 2), from the first observation in 1953) [27], the hallmark of NLSD, found in 100% of patients, in different percentage of leucocytes (from 10 to 100%) in relation to the residual function of the mutated enzymes [25]. Once patients with the test of Jordans’ anomaly are selected, the confirmation arrives from the research of genetic mutations in two different genes: *PNPLA2* and *CGI-58* [2,3] The profile of acylcarnitine and carnitine is typically normal.

#### 2.2.3. Therapy

No effective treatment exists to date for NLSD. No therapeutic trial with significant numbers of patients and long term follow up has been conducted. In the era previous to genetic tests, medium-chain triglyceride diet and oral carnitine has been tested in some individual with improvement of clinical and instrumental findings [28]. Improvement with PPAR-α has been published in the last years, both in vitro and in vivo. Experiments in an animal model showed that PPAR-α agonists completely reverses the mitochondrial defects, restores normal heart function and prevents premature death [29]. Peroxisome proliferator-activated receptors (PPARs) are nuclear hormone receptors that regulate genes involved in energy metabolism and inflammation. Reconstituting normal PPAR target gene expression by pharmacological treatment of ATGL-deficient mice with PPAR-α agonists completely reverses the mitochondrial defects, restores normal heart function and prevents premature death. These findings reveal a potential treatment for the excessive cardiac lipid accumulation and often-lethal cardiomyopathy in people with neutral lipid storage disease, a disease marked by reduced or absent ATGL activity [28]. Bezafibrate have been tested in sporadic cases in humans with encouraging but not definitive results [29]. Beta-adrenergic drugs have been tested in cultured fibroblasts with some reduction of cytoplasmic lipids (personal observation). Steroid improve PNPLA2 transcription in vitro [30] but failed to ameliorate stably clinical conditions. Some cases of NLSD-I have had some benefit with retinoids [31].

### 2.3. Phosphatidic Acid Phosphatase Deficiency (Lipin Deficiency)

*Lipin 1* gene (*LPIN1*) is expressed in adipose tissue and skeletal muscle. LPIN1 had phosphatase function in cytosol and with transcriptional regulatory function in nucleus, mediates in cytosol the penultimate step of triacyl glycerol (TAG) synthesis [32,33]. LPIN1 deficiency in muscle can simultaneously increase the early stages of autophagy and reduce the clearance of autophagic vacuoles. Autosomal recessive LPIN1 deficiency is one of the most common causes of severe recurrent rhabdomyolysis in childhood [34].

#### 2.3.1. Clinical Presentation

Episodic weakness with rhabdomyolysis is suspected caused by LPIN1 deficiency in particular in early age. Exercise, febrile illness, anesthesia or fasting caused crisis. The disease can cause death in 33% of cases, for hyperkalemia or cardiac arrest. Cardiomyopathy and hepatic steatosis have been detected at autopsy. The frequency of crisis decreased with age. Between episodes, patients are typically normal. Some practice regularly sport, but rarely aged patients can show stable proximal myopathy. Heterozygote patient can very rarely be symptomatic, sometimes after statins therapy.

#### 2.3.2. Laboratory Data and Diagnosis

Creatine kinase levels during crisis are >10,000 IU/L, plasma acylcarnitine levels are normal. Muscular tissue should be obtained during wellness period to exclude glycogenosis and mitochondrial respiratory chain disorders, although ragged red fibers have been described in some biopsies of LPIN1-deficient patients [33]. Sequence analysis of LPIN1 is the preferred diagnostic method. In a study of 141 patients with rhabdomyolysis, about 13% were found to have homozygous *Lipin1* mutations, 89% were aged 2–6 years; very isolated patients were diagnosed after 40 years.

#### 2.3.3. Therapy

The treatment of rhabdomyolysis in LPIN1 deficiency is symptomatic: aggressive intravenous fluid administration and monitoring of electrolytes, kidney, and cardiac functions as for episodes of rhabdomyolysis of any cause [34,35]. *Lipin 1* gene (LPIN1) mutations lead to cellular energy deficiency and cause up to 50% of the rhabdomyolysis episodes seen in pediatric patients. These episodes are associated with poor prognosis, as treatment options have been limited. Has been proposed a therapeutic strategy maintaining high caloric intake during viral infections or excessive physical activity and treating the rhabdomyolysis episodes with intravenous high-concentration glucose. This therapy limited the number and duration of rhabdomyolysis episodes [36].

### 2.4. Carnitine-Acylcarnitine Translocase (CACT) Deficiency

Carnitine-acylcarnitine translocase (CACT) deficiency is a rare autosomal recessive due to mutations in *SLC25A20* gene (OMIM 212138), belongs to the family of SLC25 mitochondrial carriers and catalyzes both unidirectional transport of carnitine and carnitine/acylcarnitine exchange in the inner mitochondrial membrane, allowing the import of long-chain fatty acids into the mitochondria where they are oxidized by the b-oxidation pathway. A bit more than 50 cases have been reported.

#### 2.4.1. Clinical Presentation

Disease onset occurs during neonatal period or early infancy and clinical picture includes heart problems (cardiac arrhythmia, cardiomyopathy, heart block), muscle weakness, seizures, abnormal liver function, and severe episodes of hypoglycemia and hyperammonemia triggered by fasting or infections. Presentations at a later age with a milder phenotype have been reported.

#### 2.4.2. Laboratory Data and Diagnosis

Biochemical study reveals hypoketotic hypoglycemia under fasting conditions, hyperammonemia, elevated creatine kinase and transaminases, dicarboxylic aciduria, very low free carnitine and abnormal acylcarnitine profile with marked elevation of the long-chain acylcarnitines.

#### 2.4.3. Therapy

Management of CACT patients consists over than fasting prevention and high-carbohydrate intake, supplementation of lipids as medium-chain triglycerides (MCT) and essential polyunsaturated fatty acids, and administration of carnitine.

### 2.5. Carnitine Palmitoyl Transferase II (CPT II) Deficiency

Carnitine Palmitoyl Transferase II Deficiency (CPT II deficiency) is a long-chain fatty-acid oxidation disorder. Long chain fatty acid through inner mitochondrial membrane requires carnitine and carnitine palmitoyl transferase. The first report appeared in 1973 [37].

#### 2.5.1. Clinical Presentation

The disease comprise three different clinical presentations: A lethal neonatal form, severe infantile form and a more common myopathic form (which can be mild with manifestations from infancy to adulthood) [38]. The two severe forms have multi-system involvement with liver failure, cardiomyopathy, seizures and early lethality. The milder, myopathic form of CPT II deficiency is the most common disorder of lipid metabolism affecting skeletal muscle but often remains under-diagnosed due to lack of understanding of its clinical presentation [39]. CPTII deficiency, in contrast with carnitine deficiency that cause a fixed myopathy, is characterized by recurrent attacks of myalgia, cramps, muscle stiffness or tenderness, transient weakness and rhabdomyolysis. The attacks are caused by fasting, high fat diet, general anesthesia, prolonged exercise, cold, emotional stress or fever. The frequency of crisis ranges from one or two attacks during the life to several attacks weekly. The myoglobinuria caused by the release of CK from necrotic muscle may lead to acute renal failure if inadequately and urgently treated. In early onset mechanical ventilation can be required if respiratory muscles are involved. Between the attacks neurological examination is completely normal.

#### 2.5.2. Laboratory Data and Diagnosis

Away from attacks the CK generally is normal. The EMG and ECG are normal. Usually liver is asymptomatic even if the ketogenesis is impaired and lipid accumulation occurs. In a study on about 50 patients with muscle CPT II deficiency crisis of myoglobinuria occurred in 86% of patients. Exercise is a frequent causative factor. The onset of disease is in 60% of patients in childhood (1–12 years). All the patients in whom biochemical activity was measured had normal enzyme activity of total CPT I + II but the activity was significantly inhibited by malonyl-CoA and Triton [40]. CPT-II may be lethal due to sudden cardiac death and may even compromise intrauterine development [41]. Muscle biopsy between the attacks is normal. During the crisis, muscle shows only nonspecific myopathic changes and isolated necrotic fibers. There is no or only slight accumulation of lipid or glycogen. Mitochondria are ultrastructurally normal. The diagnosis is suspected for clinical history and myoglobinuria. Plasma acylcarnitine analysis showed increases in C16 and C18 species with normal free carnitine. But using C16 and C18:1 concentration as indices diagnosis can missed as demonstrated in newborn screening. In a recent work has been adopted the (C16 + C18:1)/C2 ratio (cutoff 0.62) and C16 concentration (cutoff 3.0 nmol/mL) as alternative indices for CPT II deficiency such that an analysis of a dried blood specimen (DBS) collected at postnatal day five. Thereafter, positive cases were assessed by measuring the fatty acid oxidation ability of intact lymphocytes and/or CPT II activity in the lysates of lymphocytes. The diagnoses were then further confirmed by genetic analysis [42]. The p.S113L was the most frequent mutation (95%) in at least one allele. Sixty percent of index patients were homozygous for this mutation. Thirteen other mutations, all in compound heterozygote form, were also identified. Attacks were triggered by fasting in almost all the patients with truncating mutations. In contrast, fasting triggered the attacks only in one third of patients with missense mutations on both alleles. The data indicate that within the muscle form of CPT II deficiency, the various genotypes have only marginal influence on the clinical and biochemical phenotype [40]. Isolated cases of symptomatic heterozygotes are reported [43].

#### 2.5.3. Therapy

The patients with CPT II deficiency can control their attacks preventing triggering factors, and remain asymptomatic by adopting a specific lifestyle and a diet with high carbohydrate doses [44]. In the recent years, new drugs have been tested to increase fatty acid oxidation. Fibrates are a class of hypolipidemic drugs that increase HDL-level by mRNA up-regulation of many lipid-metabolism genes via the interaction with the steroid-thyroid transcription factor PPAR-α. Recent studies demonstrated that bezafibrate increases CPT2mRNA and normalizes enzyme activity in mild form of CPT II deficient cultured fibroblasts and myoblasts [40]. As in Very-Long-Chain acyl-CoA Dehydrogenase Deficiency (VLCADD), Long-Chain 3-hydroxyacyl-CoA Dehydrogenase Deficiency (LCHADD), Multiple Acyl-CoA Dehydrogenase Deficiency (MADD), or CACT deficiency, also in CPT2 deficiency urgent treatment should be meticulous as there is a high risk of serious complications. The major complications are encephalopathy, cardiomyopathy, hypoglycemia and rhabdomyolysis. The treatment if the patient is unwell or vomiting and hypoglycemic is glucose 200 mg/kg at once. If the conditions are relatively good, medication as carnitine (100 mg/kg at once) and feeds may be given orally, in line with British Inherited Metabolic Diseases Group recommendations. Glucose infusion is of benefit during or after myoglobinuria, but oral glucose does not achieve the same effects, probably because of its low penetration in muscle and because of the insulin-response, which inhibits muscle glycogenolysis [44]. Recently, a clinical trial with bezafibrate in CPT II deficiency did not show any efficacy [45]. It is important to avoid drugs including non-steroidal anti-inflammatory drugs (as ibuprofen, often prescribed for post-exercise muscle pain because may increase the risk of renal failure). Management of disease is mainly oriented to prevent renal failure during a crisis and involves avoidance of known triggers, reducing dietary long-chain fats, providing carnitine and adequate hydration. It is possible to recommend extra carbohydrate intake before and during sustained exercise and frequent meals.

### 2.6. Very-Long-Chain Acyl-CoA Dehydrogenase (VLCAD) Deficiency

Very-Long-Chain Acyl-CoA Dehydrogenase (VLCAD) is a recessive disorder due to variants in *ACADVL* gene. Most of mutations are privates with strong genotype-phenotype correlation [46,47,48]. Long-Chain Acyl-CoA Dehydrogenase (LCAD) deficiency is coded by Acyl-CoA Dehydrogenase Long Chain (*ACADL*) gene. Even if the pathophysiological mechanism of disease has not yet been fully elucidated, likewise in other long-chain FAO defects, detergent-like action on muscle membranes due to a toxic accumulation of long-chain fatty acids and long-chain acylcarnitines, has been suggested [48].

#### 2.6.1. Clinical Presentation

LCAD and VLCAD share a similar clinical presentation and a number of less recently reported LCAD cases were later believed to be attributable to VLCAD [14,48]. VLCAD most often manifests as with cardiac or liver involvement in childhood and recurrent rhabdomyolysis in adults [50,51]. Crisis of rhabdomyolysis could be triggered by prolonged exercise, fever, cold or fasting. Differential diagnosis should be made with CPT II and lipin1-related early-onset rhabdomyolysis. 

#### 2.6.2. Laboratory Data and Diagnosis

Muscle biopsy shows mild lipid accumulation, mainly in type 1 fibers in among 30% of patients [50]. Occasionally mild mitochondrial proliferation with reduced mitochondrial chains activity has been reported. Biochemical analysis could reveal abnormal acyl carnitines profile with predominant C_14:1_ chain. Interestingly, increased acylcarnitine profile suggestive of VLCAD can be occasional found in false positive and asymptomatic carriers, as shown in a recent new born screening study [52]. Diagnosis should be confirmed by residual enzymatic activity in lymphocytes (<12%) and molecular analysis of *ACADVL* gene. MRI studies may show specific alteration patterns, with short tau inversion recovery (STIR) weighted signal intensity increase in almost all muscle groups [50].

#### 2.6.3. Therapy

Beyond general recommendations, specific treatments for VLCAD deficiency are not yet available, even if three patients have been successfully treated with triheptanoin acid implementation which led to rapid clinical improvement of muscle weakness and cardiomyopathy [53] and medium chain triglycerides (MCT) oil supplementation has been reported as effective in VLCAD-related cardiomyopathy [54]. In a double-blind randomized crossover study of bezafibrate. in five individuals with VLCAD deficiency no improvement could be detected [44]. Recently promising results have been reached by rAAV9-mediated gene therapy approach in mice [54].

### 2.7. Acyl-CoA Dehydrogenase 9 (ACAD9) Deficiency

Acyl-CoA Dehydrogenases (ACADs) belong to a family of flavoenzymes involved in the beta-oxidation of acyl-CoA and amino acid catabolism. Acyl-CoA Dehydrogenase 9 (ACAD9) is most homologous (65% amino acid similarity) to VLCAD [55]. Despite a significant overlap of substrate specificity, it appears that ACAD9 and VLCAD are unable to compensate for each other in patients with either deficiency. This support the presence of two independently regulated functional pathways for long-chain fat metabolism, indicating that these two enzymes are likely to be involved in different physiological functions. Both ACAD9 and VLCAD function as homodimers associated with the inner mitochondrial membrane and catalyze the initial step of the fatty acid oxidation (FAO) [56]. ACAD9 mutations impair the complex I assembly and activity resulting in complex I deficiency, without disturbing long-chain fatty acid oxidation [57].

#### 2.7.1. Clinical Presentation

ACAD9 deficiency is dominated by cardiomyopathy (85%), muscular weakness (75%) and exercise intolerance (72%). Severe intellectual deficits and severe developmental delays in few patients. Other features are lactic acidosis, myopathy and developmental delay. Age of onset, severity of symptoms and progression are variable [58]. 

#### 2.7.2. Laboratory Data and Diagnosis

Diagnosis can be confirmed by both biochemical and genetic tests [58]. 

#### 2.7.3. Therapy

There is only one report about patients, with a predominance of myopathic features, that showed alleviation of symptoms under riboflavin treatment [59].

### 2.8. Medium-Chain Acyl-CoA Dehydrogenase (MCAD) Deficiency

Medium-Chain Acyl-CoA Dehydrogenase (MCAD) deficiency is the most frequent FAO disorder with an incidence around 1:9000 live births [60,61]. It is a recessive disorder caused by mutations in *ACADM* gene. Variant c.985A > G (p.K329E) is the most frequent mutation among populations of European descent, representing 54–90% of disease alleles, with homozygotes representing about 47–80% of MCAD patients [62]. Conversely in Japan, K329E is not found and the most prevalent mutation is c.449_452delCTGA (p.T150Rfs), followed by R17H, G362E, R53C and R281S which account together for approximately 60% of all mutated alleles [63].

#### 2.8.1. Clinical Presentation

MCAD typically manifests during infancy with reversible episodes of hypoketotic hypoglycemia and Reye-like syndrome, frequently accompanying by vomiting and trouble of vigilance, which can lead to coma and death in unrecognized and untreated patients. MCAD deficiency has been also associated to many cases of sudden infantile death. Nevertheless, later onset and exercise or alcohol-induced episodes of rhabdomyolysis have also been reported [64]. Chronic skeletal muscle weakness is less common than in other FAO disorders [65]. Maternal complications in pregnancy are not limited to carriers of LCHAD mutation and infant patients with deficiencies of CPT I, MCAD, and Short-Chain Acyl CoA Dehydrogenase Deficiency (SCAD) were also born to mothers who developed liver disease during their pregnancies [66,67,68].

#### 2.8.2. Laboratory Data and Diagnosis

MCAD is confirmed by specific findings by acylcarnitine profile with increased C_8_ and C_10:1_, as well as their corresponding free fatty acids are also detectable in high amounts in urines. Dicarboxylic aciduria, with abnormal acyl glycine excretion could also occur. 

#### 2.8.3. Therapy

Beneficial L-carnitine implementation in MCAD remains controversial [69,70]. Breast feeding and avoidance of fasting with prompt institution of glucose supplementation remain the mainstay of therapy.

### 2.9. Short-Chain Acyl-CoA Dehydrogenase (SCAD) Deficiency

It is due to recessive mutations in *ACADS* gene, although genetic characterization of these patients remains difficult [71,72,73]. No patients with null mutation on both alleles have been identified as it would be probably incompatible with life, and to differentiate between benign variants and disease susceptibility alleles remains challenging. In fact, patients with SCAD deficiency frequently have compound heterozygotes for rare mutations and common variant alleles which are present in high percentage of general population and could represent a genetic susceptibility polymorphism.

#### 2.9.1. Clinical Presentation

Clinical manifestations may be triggered by genetic or environmental factors [74,75]. A range of symptoms have been reported, with possible fatal metabolic decompensation, including (1) intermittent metabolic acidosis and neonatal hyperammonemia; (2) infantile onset lipid storage myopathy with failure to thrive, developmental delay, hypotonia, seizures, and hyperreflexia [71]; (3) late onset progressive myopathy and ophthalmoplegia with multiminicore myopathy [72]; and (4) asymptomatic individuals [73]. More recently a large cohort study of 114 patients between 0 and 50 years with SCAD deficiency demonstrated that 25% of patients manifests some clinical symptoms on the first day of life, 61% within the first year of life, and only 4% after than 10 years. Patients were grouped in 3 main groups accordingly with prominent symptoms: (1) failure to thrive with feeding difficulties and hypotonia as the most characteristic features (20% of patients): (2) developmental delay and seizures (25% of patients); (3) developmental delay and hypotonia without seizures (30% of patients). A fourth symptomatic group consisting of 14% showed failure to thrive, developmental delay, and hypotonia. The remaining 7% had a heterogeneous mixture of other symptoms including dysmorphic features, myopathy, cardiomyopathy, hepatic steatosis, respiratory distress, and intrauterine growth retardation. Finally 4% of patients have been reported to have no symptoms [74].

#### 2.9.2. Laboratory Data and Diagnosis

The major biochemical hallmark is the presence of increased concentrations of ethylmalonic and methyl succinic acids in urine, other than adipic, suberic, sebacic acids. This finding could also be due to the presence of one of two relatively common variants of SCAD (G625A and C511T) which predispose to excessive ethylmalonic acid production and are also present in 14% of general populations either in homozygous or double heterozygous form [75]. Plasma acylcarnitine profile presents a high C_4_ species.

#### 2.9.3. Therapy

No specific treatments are currently available.

### 2.10. Mitochondrial Trifunctional Protein (MTP) Deficiency

The Mitochondrial Trifunctional Protein (MTP) is a heterocomplex of four alpha and four beta subunits, which are encoded by two nuclear genes (*ADHA* and *ADHB* respectively) and catalyze 3 steps of mitochondrial beta-oxidation: (1) the Long-chain enoyl-CoA hydratase (ECH) and (2) long-chain L-3-hydroxyacyl-CoA dehydrogenase (LCHAD) activities by alpha subunits, and (3) the long-chain 3-ketoacyl-CoA thiolase (LCKAT) activity by beta subunits [76,77,78]. Two different biochemical phenotypes of MTP deficiency are possible [79,80]. The first includes deficiency of all three enzyme activities of the MTP, with loss of both the alpha and beta subunits by immunoblotting studies. The second has isolated LCHAD deficiency with normal amounts of the alpha and beta subunits. Even if isolated LCHAD seemed to be more frequent than MTP, many cases previously reported as LCHAD are now known to be MTP deficiency [81].

#### 2.10.1. Clinical Presentation

There are three main clinical presentation of MTP/LCHAD deficiency with variable overlap [82]: (1) severe neonatal with encephalopathy and hepatopathy possibly manifesting with Reye-like syndrome associated to high mortality rate and sudden death [83], which could also lead to a severe maternal illness occurring during pregnancy of affected fetus, including (a) acute fatty liver pregnancy (AFLP) syndrome, (b) hypertension or hemolysis, elevated liver enzymes, and low platelets (HELLP) syndrome and (c) hyperemesis gravidum [84,85]; (2) early onset with cardiomyopathy and recurrent rhabdomyolysis [86]; (3) milder late-onset myo-neuropathic form characterized by recurrent rhabdomyolysis often associated to respiratory failure, optic retinopathy and sensorimotor axonal neuropathy [87]. The sensorimotor neuropathy is a helpful feature to distinguish MTP among other FAO defects and sometimes may mimic the spinal muscular atrophy (SMA) phenotype [88].

#### 2.10.2. Laboratory Data and Diagnosis

Muscle biopsy may have a neurogenic appearance with angulated atrophic fibers secondary to neuropathic involvement with type 1 fiber predominance, and lipid accumulation has been only occasionally reported. In LCHAD specific deficiency it may be found and impaired respiratory chain activity (complexes I, II, III) and RRF [89,90]. As other FAO defects, total carnitine can be decreased and tandem mass spectrometry inconstantly show elevation of 3OH-C_16_, 3OH-C_18_, C_18:1_, C_18:2_, especially during rhabdomyolysis episodes, whereas during stable clinical conditions, acyl carnitine profile can be normal. Study of FAO defect in fibroblast could show a residual activity of PAL^C14^ > MYR^H13^ > PAL^H3^. MRI contribute to the diagnosis, showing increase of T1W and STIR signal intensity [10].

#### 2.10.3. Therapy

Specific treatments have been proposed for MTP/LCHAD: docosahexaenoic acid (DHA) supplementation (<20 kg: 60 mg/day; >20 kg: 120 mg/day in children) [91,92], Triheptanoin [53] and peroxisome proliferators activated receptor (PPAR) agonists [93,94].

### 2.11. Short-Chain L-3-Hydroxyacyl-CoA Dehydrogenase (SCHAD) Deficiency

3-Hydroxyacyl-CoA Dehydrogenase (HAD) is coded by *HADH* (or *HADHSC*) gene and catalyses the oxidation of straight chain 3-hydroxyacyl-CoAs in mitochondrial FAO [95]. HAD has a preference for medium chain substrates, whereas short chain 3-hydroxyacyl-CoA dehydrogenase also known as 17beta-hydroxysteroid dehydrogenase type 10 (coded by *SCHAD* gene on chromosome X), acts on a wide spectrum of substrates, including steroids, cholic acids, and fatty acids, with a preference for short chain methyl-branched acyl-CoAs. SCHAD, also known as 17 beta-hydroxysteroid dehydrogenase type 10, is important in brain development and aging. So, SCHAD is not a member of the HAD family but belongs to the short chain dehydrogenase/reductase superfamily. It has been known that previously reported cases of *SCHAD* deficiency are instead due to an inherited *HADH* deficiency [96]. M/SCHAD deficiency has been only rarely reported. 

#### 2.11.1. Clinical Presentation

Different clinical phenotypes have been reported including: (1) recurrent myoglobinuria, myopathy, cardiomyopathy, and hypoketotic hypoglycemic encephalopathy [97]; (2) fasting-induced vomiting, ketosis and hypoglicemia [98]; (3) fatal infantile hepatic involvement and steatosis [99]; (4) Hyperinsulinism with hypoglycemia has also been reported [100].

#### 2.11.2. Laboratory Data and Diagnosis

Organic aciduria is reported and the sequencing of the HAD gene led to diagnosis [97].

#### 2.11.3. Therapy

No therapy is reported.

### 2.12. Medium-Chain 3-Ketoacyl-CoA Thiolase (MCKAT)

Only one Japanese male neonate who died at 13 days of age has been described [101].

#### 2.12.1. Clinical Presentation

The patient presented at 2 days of age with vomiting, dehydration, metabolic acidosis, liver dysfunction, and rhabdomyolysis with myoglobinuria.

#### 2.12.2. Laboratory Data

Biochemical study revealed markedly elevated excretion of lactic acid, 3-hydroxybutyric acid, and saturated and unsaturated C6–C16 dicarboxylic acids, with predominant C12–C16 species. Immunoprecipitation with antibodies raised against medium chain 3-ketoacyl-CoA thiolase revealed a 60% decrease compared with controls.

#### 2.12.3. Therapy

No therapy is reported.

### 2.13. Multiple Acyl-CoA Dehydrogenase Deficiency (MADD)

Multiple acyl-CoA Dehydrogenase Deficiency or Glutaric aciduria type II (GAII) is a recessive disorder classically associated to defect in one of the two electron flavoprotein transporters implicated in electron transfer from acyl-CoA dehydrogenases to the respiratory chain: (1) ElectronTransfer Flavoprotein (ETF, coded by *ETFA* and *ETFB* genes) and (2) ETF-ubiQuinone Oxidoreductase (ETFQO, coded by *ETFDH* gene) [102].

#### 2.13.1. Clinical Presentation

MADD could manifest with a wide range of clinical presentations which have been divided into three main groups: (1) neonatal onset with congenital anomalies, (2) neonatal onset without anomalies, and (3) mild or late onset. The first two groups represent the severe form of disease whereas the last group represents the mild form.

Neonatal-onset with congenital anomalies are often premature, hypotonic and have a characteristic smell like sweaty feet. Severe hypoglycemia frequently manifests within the first 24–48 h of life. Hepatomegaly is frequently present with metabolic acidosis. Complete clinical picture could include macrocephaly, pachygyria and cortical heterotopias, enlarged kidneys, facial dysmorphism, congenital vertical talus, muscular defects of the anterior abdominal wall, and anomalies of the external genitalia [103]. Most of these patients die within the first week of life. The second group of infants without congenital anomalies usually develop hypotonia, hepatomegaly, hypoglycemia, metabolic acidosis, and a sweaty foot smell within the first few days of life. Early treated patients die within a few months because of severe cardiomyopathy. Infants manifesting with hypoglycemic crisis and develop later typical Reye-like episodes and have survived longer. Late onset GA II is highly variable in clinical presentations ranging from asymptomatic adults with exertional rhabdomyolysis to proximal myopathy often with hepatomegaly and episodic metabolic crisis of vomiting, hypoglycemia, and acidosis which could be lethal. Given the wide spectrum of clinical presentation, late-onset MADD is probably an underdiagnosed condition and clinical awareness among physicians is warranted [104]. Recently, four main clinical presentations have been proposed for adult onset: (1) exercise intolerance (2) isolated muscle weakness in one case (rare); (3) multisystemic with acute central nervous system (CNS) impairment or liver dysfunction; (4) pseudo-myasthenic presentation with fluctuating symptoms [105]. Cardiomyopathy and sudden death have also been reported in both neonatal and later onset [106,107]. Finally, isolated myopathic Coenzyme Q10 deficiency caused by mutations in the *ETFDH* has been reported [108]. Patients show proximal myopathy, exercise intolerance, CK elevation, lipid storage at muscle biopsy, reduced activities of mitochondrial complexes and decreased CoQ10 in muscle. Late-onset GAII and the myopathic form of CoQ10 deficiency could be allelic diseases. Studies of genotype-phenotype correlation showed that homozygous mutations in *ETFA* and *ETFB* seem to manifest more frequently with severe neonatal onset while mutations in *ETFDH* with residual activity often present later onset [109].

#### 2.13.2. Laboratory Data and Diagnosis

Muscle biopsy finding in MADD is characterized by a reversible massive lipid droplets accumulation with occasional Ragged Red Fibers (RRF) and COX negative fibers. Biochemical analysis reveals a low concentration of total carnitine with elevated concentration of all length acylcarnitines (C_4_–C_18_). Urinary organic acids typically show dicarboxylic aciduria (C_5_–C_10_) and acylglycine derivates. Reduced activity of respiratory chain complexes I, II, III and IV on muscle biopsy with reduction of CoQ10 level have been reported in *ETFDH*-related cases of CoQ10 deficiency [108]. Diagnosis confirmation is achieved by molecular analysis showing recessive mutations in *ETFDA, ETFDB* or *ETFDH* genes [110,111]. Nevertheless, some of cases remain genetically unexplained. Interestingly, the last few years a number of riboflavin transporter genes (*SLC52A1, SLC52A2, SLC52A3*) [112,113,114], mitochondrial flavin adenine dinucleotide (FAD) transporter gene (*SLC25A32*) [115] and FAD synthetase gene (*FLAD1*) [116] have been linked to undiagnosed cases of riboflavin-responsive MADD. In this context, MADD could be considered as a group of disorders instead of a single entity.

#### 2.13.3. Therapy

MADD can be responsive to riboflavin supplementation (50–100 mg 3 times daily) and to low fat diet avoiding long fasting periods can also be helpful [110,111]. This could be due to a stabilizing which improves ETF conformation, leading to a more stable and active enzyme [110].

## 3. Acquired Disorders of Lipid Metabolism in Muscle

### 3.1. HIV

A syndrome of lipodystrophy, with subcutaneous fat loss, has been seen in patients with HIV [117,118], when treated with a combination of antiretroviral therapy, that causes adipose tissue dysfunction and is associated with impaired fat oxidation, lipid accumulation in muscle and liver, and mitochondrial dysfunction in skeletal muscle leading to dyslipidemia and insulin resistance [118]. The sphingolipids, especially ceramides, and DiAcylGlycerol (DAG), have been supposed to induce insulin resistance. Besides antagonize insulin signaling recent data also shows that ceramides impair mitochondrial functions. Patients with HIV-infection and lipodystrophy are characterized by lipid and glucose disturbances and increased levels of circulating IL-18. Recently was found that IL-18 mRNA expression was lower in skeletal muscle in patients with HIV-lipodystrophy and that IL-18 was implicated in lipid metabolism [118].

### 3.2. Alcohol

Chronic myopathy, characterized by progressive weakness in proximal muscles with difficulty climbing stairs, walking, and rising from squatted and seated positions [119] occurs in 33% to 67% of alcoholics [120]. Muscle biopsy shows atrophy of glycolytic (Type 2) myofibers, isolated myofibers with moth-eaten appearances in sections stained for oxidative enzyme activity. Progressive myofiber atrophy leads to wasting with up to 30% loss of muscle mass [121]. Ultrastructural studies revealed increased lipid deposition, dilatation of the sarcoplasmic reticulum, loss of myofilaments [122] and anomalies of mitochondrial cristae. Myopathy caused by chronic ethanol exposure seems to be related to reduced mRNA expression of insulin-like growth factor 1 (IGF-1) polypeptide, insulin, IGF-1, and IGF-2 receptors, insulin receptor substrate (IRS) 1, and IRS-2 [123].

### 3.3. Drugs

Bortezomib, treatment for multiple myeloma, in 30% of treated patients can cause reversible metabolic myopathy, pathologically characterized by excessive storage of lipid droplets with mitochondrial abnormalities. In human myoblasts bortezomib at low concentrations leads to excessive storage of lipid droplets with mitochondrial abnormalities [124].

## 4. Conclusions

Lipid myopathies are phenotypically and genotypically heterogeneous. The knowledge of clinical presentations of these diseases is important for deducing the diagnosis. Their acute and severe symptoms in internships and first aid departments, can be susceptible to life-saving treatment. An accurate observation of clinical symptoms combined with humoral and instrumental diagnostic approach can lead a diagnosis of metabolic myopathy. In addition, genetic analyses are generally helpful for an accurate and specific diagnosis when the clinical phenotype is not clear and laboratory tests are not definitive. In the future, the studies on muscle lipid defects should be directed to the research of novel causative genes and to the comprehension of the pathogenesis of these rare and disabling diseases, improving the knowledge of lipid metabolism and leading to new therapeutic strategies.

## Figures and Tables

**Figure 1 jcm-07-00472-f001:**
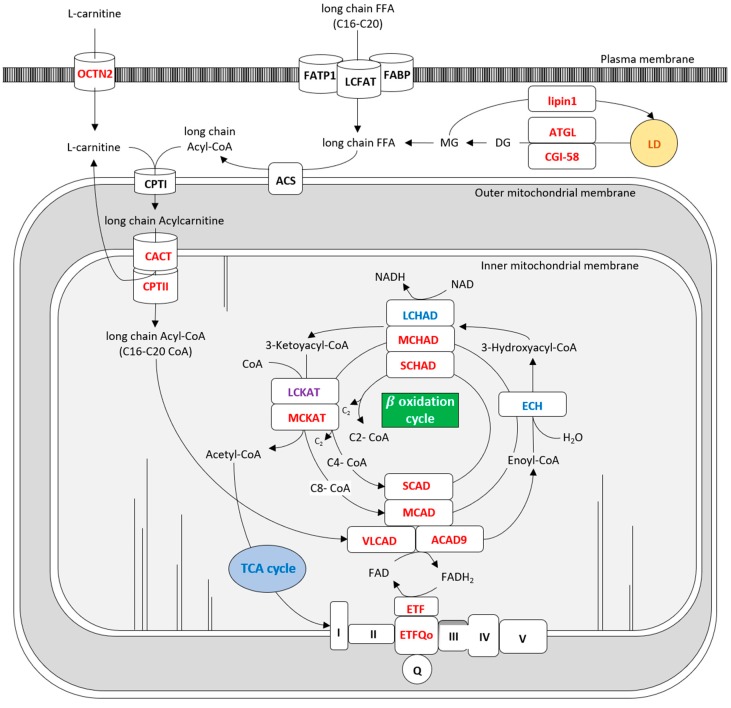
Metabolic pathway of lipid metabolism in muscle. ACDA9: Acyl-CoA dehydrogenase 9; ACS: Acetyl-CoA synthetase; ATGL: Adipose triglyceride lipase; CACT: Carnitine-acylcarnitine translocase; CGI58: Comparative gene identification-58; CPT I: Carnitine palmitoyl transferase I; CPT II: Carnitine palmitoyl transferase II; DECR: Dienoyl-CoA reductase; DG: Diglycerides; ECH: Enoyl-CoA hydratase; ETF: Electron transport flavoprotein; FABP: Fatty acid binding protein; FAD: flavin adenine dinucleotide; FADH: flavin adenine dinucleotide reduced; FATP1: Fatty acid transport protein 1; FFA: Free fatty acid; LCFAT: Long chain fatty acid; LCHAD: long-chain 3-hydroxyacyl-CoA dehydrogenase; LCKAT: long-chain 3-ketoacyl-CoA thiolase; LD: Lipid droplet; Lipin1-Lipin 1; MCAD: Medium-chain acyl-CoA dehydrogenase; MG: Monoglycerides; NAD: Nicotinamide adenin dinucleotide; NADH: Nicotinamide adenin dinucleotide reduced; OCTN2: Organic Cation/Carnitine Transporter 2; Q: Coenzyme Q; SCAD: Short-chain acyl-CoA dehydrogenase; VLCAD: Very-long-chain acyl-CoA dehydrogenase; TCA: tricarboxylic acid.

**Figure 2 jcm-07-00472-f002:**
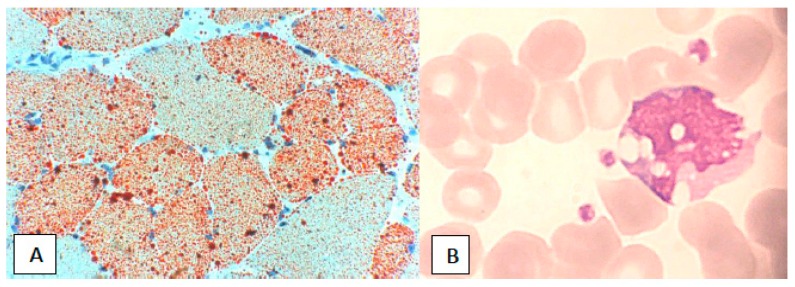
Massive lipidosis and Jordans’ anomaly. (**A**) Lipid storage in muscle fibers from NLSD-M patient; optic microscopy, Oil-red-O (ORO) stain, 20×. (**B**) Leucocyte presenting lipid droplets (Jordans’ anomaly); optic microscopy, Giemsa, 100×.

**Table 1 jcm-07-00472-t001:** Lipid myopathies overview.

		Disease Name	Gene	Protein	Predominant Symptoms	Laboratory Diagnosis	Lipid Storage	Therapy
PM	LMI	Primary Carnitine Deficiency (**PCD**)	*SLC22A5*	OCTN2	C, E, Hy, M, R, RL	LC, AC	+++	Carnitine, preventing trigger factors
Cytoplasm	LMII	Neutral lipid storage disease type M (**NLSD-M**)	*PNPLA2*	ATGL	C, IOL, M	JA, NC, CK	+++	Low fat diet, MCT, exercise, fibrates, beta-adrenegic drugs
	LMII	Neutral lipid storage disease type I (**NLSD-I**)	*CGI58/ABHD5*	CGI58	He, Ic, M	JA, NC	+++	Low fat diet, MCT, carnitine, triheptanoin acid, acitrein
	LMIV	Phosphatidic acid phosphatase deficiency (**Lipin Deficiency**)	*LPIN*	Lipin 1	C, He, M, R	CK, NC	+/−	Glucose and fluids infusion, monitoring of vital function
	LMV	Carnitine-Acylcarnitine Translocase) Deficiency (**CACT**)	*SLC25A20*	CACT	C, E, He, Hy, M	OA, LC, AC	NR	Avoid fasting, high carbohydrate intake, MCT, polyunsaturated fatty acids, carnitine.
Mitochondrion	LMVI	Carnitine palmitoyl transferase II deficiency (**CPT II**)	*CPTII*	CPTII	M, My, R, TM	AC, NC, DBS	+/−	Glucose infusion, carnitine, avoid FANS, preventing trigger factors
	LMVII	Very-long-chain acyl-CoA dehydrogenase deficiency (**VLCAD**)	*ACADVL*	VLCAD	C, He, R	AC	+/−	Triheptanoin acid, MCT, N-acetylcisteine, avoid fasting,
	LMVII	Acyl-CoA dehydrogenase 9 deficiency (**ACAD9**)	*ACAD9*	ACAD9	C, Ex, Hy, He, R, RL		−	Riboflavin
	LMIX	Long-chain acyl-coA dehydrogenase (**LCAD**)	*ACADL*	LCAD	C, He, R	AC	+/−	Riboflavin
	LMX	Medium-chain acyl-CoA dehydrogenase deficiency (**MCAD**)	*ACADM*	MCAD	C, Ex, He, Hy, M, R	AC	−	Avoid fasting, glucose infusion
	LMXI	Short-chain acyl-CoA dehydrogenase deficiency (**SCAD**)	*ACADS*	SCAD	C, Ex, MA, M, no Hy	AC, OA	+/−	Low fat diet, carnitine, riboflavin
	LMXII	Mitochondrial trifunctional protein deficiency (**MTP**)	*ADHA*	ECH, LCHAD	C, E, He, M, N, NH, R, RL	AC	+/−	Decosahenoxic acid (DHA), Triheptanoin acid, PPAR
			*ADHB*	LCKAT		AC	−	
	LMXIII	Short-chain L-3-hydroxyacyl-CoA dehydrogenase deficiency (**SCHAD**)	*HADH*	SCHAD	C, M, He, Hy, R	OA	+/−	No therapy reported
	LMXIV	Medium-chain 3-ketoacyl-CoA thiolase (**MCKAT**)	*MCKAT*	MCKAT	He, MA, R	OA, AC	NR	No therapy reported
	LMXV	Multiple acyl-CoA dehydrogenase deficiency (**MADD**)	*ETFA, ETFB*	ETF	C, D, Ex, He, Hy	LC, AC	+/−	Riboflavin, low fat diet, avoid fasting, CoQ
			*ETFDH*	ETFQO	C, D, Ex, He, Hy	LC, AC	+++	Riboflavin, low fat diet, avoid fasting, CoQ

PM: plasma membrane; LM: lipid myopathy; C: cardiomyopathy; E: encephalopathy; Ex: excercise intolerance; D: dysmorphims; He: hepatopathy;; Hy: hypoglycemia; M: fixed myopathy; My: myalgia; MA: metabolic acidosis; N: neuropathy; NC: normal carnitine; NH: neonatal hypotonia; OA: organic aciduria; R rabdomyolisis; RL: Reye like syndrome; Ic: ichthiosis; IOL: internal organ lipidosis; TM: transient myopathy; AC: abnormal acylcarnitine; EL: elevated lactate; CK: creatine kinase elevation; JA: Jordans’ anomaly; LC: low carnitine; DBS: dried blood spot; MCT: medium chain triglycerides; CoQ: coenzyme Q; +++: abundant; +/−: mild or absent; NR: not reported.

**Table 2 jcm-07-00472-t002:** Secondary Carnitine Deficiency *.

Genetically Determined Metabolic Errors	Acquired Medical Conditions
*Increased Esterification due to acyl-CoA Accumulation* CACT deficiency CPT II deficiency VLCAD LCAD LCHAD MTP MCAD SCAD SCHAD MADD Beta-Hydroxy-beta-methylglutaryl-CoA lyase deficiency Isovaleric acidemia Propionic acidemia Methylmalonic aciduria Glutaryl-CoA dehydrogenase deficiency (glutaric aciduria type I) β-ketothiolase deficiency *Decreased Biosynthesis* Homocystinuria 5-mtetrahydrofolate reductase deficiency Adenosine deaminase deficiency Ornithine transcarbamylase heterozygote state *Increased Urinary Loss* Cystinosis Cytochrome oxidase deficiency	*Decreased Biosynthesis* Cirrhosis Chronic renal disease Extreme prematurity *Dietary Deficiency* Chronic TPN without carnitine supplementation Malabsorption (cystic fibrosis, short-gut syndrome) Unsupplemented Soybean protein-derived infant formula Decreased Body Stores Extreme prematurity Intrauterine growth retardation Infant of carnitine-deficient mother Increased Urinary Loss Fanconi syndrome Renal tubular acidosis **Iatrogenic Factors** Increased Esterification and competitive inhibition of carnitine uptake by valproylcarnitine Chronic valproic acid administration Impaired Hepatic Biosynthesis Chronic valproic acid administration Increased Loss Chronic hemodialysis

CACT: Carnitine-Acylcarnitine Translocase Deficiency; CPT II: Carnitine Palmitoyl Transferase II Deficiency; VLCAD: Very-Long-Chain acyl-CoA Dehydrogenase Deficiency; LCAD: Long-chain acyl-coA dehydrogenase; MTP: Mitochondrial Trifunctional Protein Deficiency; MCAD: Medium-Chain Acyl-CoA Dehydrogenase Deficiency; SCAD: Short-Chain Acyl-CoA Dehydrogenase Deficiency; SCHAD: Short-Chain L-3-Hydroxyacyl-CoA Dehydrogenase Deficiency; MADD: Multiple Acyl-CoA Dehydrogenase Deficiency. * modified from Tein et al. [20].

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
