# Peer review of "Lipid Myopathies"

_jcm, 2018, doi:10.3390/jcm7120472_

Reviewer 1 Report

The paper “Lipid Myopathies (LM)” by Pennisi, Garibaldi and Antonini is a comprehensive overview essentially focused on the classification of LM on the basis of gene mutation, description of related-impaired lipid beta oxidation, diagnosis and therapies.

I would suggest some minor changes in order to help the readers.

n particular:

-          An initial general description of LM at the beginning of the Introduction would be desirable;

-          The description of the main general/physiological biomolecular processes regulating lipid metabolism should be summarized in a dedicated subparagraph and not inserted within in the subparagraphs dedicated to different disease subtypes (i.e., the sentence from lane from 169 to 173, par. 3.2. NLSD)

-          Addition of the main related references would complete info shown in Tab 1;

-          Authors could address comments on ongoing trials;

-          Some typos and errors should be corrected; words should be consistently written (i.e., beta oxidation/B oxidation/β oxidation/b oxidation; the same for PPARalfa/PPAR-α)

-          The recent review on LM doi.org/10.1016/j.plipres.2018.08.001 should be commented and quoted.

Author Response

Here attached the response to reviewer 1.

Reviewer 2 Report

The manuscript”Lipid Myopathies” is a review of the muscular lipid myopathies. Generally the manuscript is well-written, and I only have minor comments:

You may want to subdivide your paragraphs, so you have a title for genetics, clinical presentation, paraclinical findings, treatment and differential diagnosis.

Some of the sentences are difficult to understand, and should be revised, generally genes are written in italic (kursive), and last there are a few typographical errors that needs correction:

This sentence: also including some those disorders of 44 FAO with muscular symptoms not included among previous lipid myopathies on line 44-45  does not make sence, please revise. The same with: The sole alteration of FA entrance machinery is able to 64 modulate FA oxidation rate indicating a high level of the metabolic regulation line 64-65.

Line 69 please changes “in” to: into the

Line 70 please change in B-oxidation to “the β-oxidation or Beta-oxidation

Line 72, what organelles? Please be more specific and revise this sentence.

Line 75, please introduce CGI58, and write in italic if it’s a gene.

Line 82, please change to red brown colored urine.

Line 85, what do you mean with deficiency in the diet?

Line 85,86, please revise.

Please spell out ORO before you use the abbreviation, and explain what the coloring shows in the muscle biopsy.

Line 91, 92, please add reference.

Line 101. What specific myopathies, please explain.

Line 129, 130, please revise, it is unclear what your subdivision is. Do you refer to line 67-70, if so please be more specific?

Please add reference to all the recommended therapies in table 1, and delete those that are not based on evidence.

Line 174, change “his” to “its”

Line 174, 175, change “MIM” to “OMIM”

Line, 181, I think it is called Arabian Peninsula

Line 183, change “basing” to “based”

Line 188, delete underscore before “in”

Line 209, who does “his” refer to? Please revise.

Line 213, genes should be in italic

Line 225-228, please add reference.

Line 256, please change “if” to “of”

Line 276, please explain why you consider CACT a myopathy?

Line 308, CPT II patients indeed have exercise intolerance, please re-write.

Line 312, 313, are you referring to the neonatal form, or do you have experience with adult CPT II patients having affected respiratory muscles. Please elaborate, and add relevant references.

Line 367, please explain what private means in this sentence.

Line 378, please add reference.

Line 390, please add reference.

Please explain why you consider Acyl-CoA dehydrogenase 9 (ACAD9) deficiency, SCAD and DECR as a lipid myopathies. This can be added to the introduction.

Line 424, Reye like syndrome?

Line 424, pleas change “ad” to “and”

Line 425, please revise

Please explain why you consider as a lipid myopathy

Line 552, please revise

Line 613, diacylglycerol

 Author Response

Here attached the response to reviewer 2.
